

# Space-filling and benthic competition on coral reefs

Emma E. George[1,2,*], James A. Mullinix[3,4,5,*], Fanwei Meng[3], Barbara A. Bailey[3,5], Clinton Edwards[6], Ben Felts[3,5], Andreas F. Haas[7], Aaron C. Hartmann[1,8,13], Benjamin Mueller[9,10], Ty N.F. Roach[1,5,11], Peter Salamon[3,5], Cynthia Silveira[1,5,12], Mark J.A. Vermeij[9,10], Forest Rohwer[1,5] and Antoni Luque[3,4,5]

[1] Department of Biology, San Diego State University, San Diego, CA, United States of America
[2] Department of Botany, University of British Columbia, Vancouver, British Columbia, Canada
[3] Department of Mathematics and Statistics, San Diego State University, San Diego, CA, United States of America
[4] Computational Science Research Center, San Diego State University, San Diego, CA, United States of America
[5] Viral Information Institute, San Diego State University, San Diego, CA, United States of America
[6] Scripps Institution of Oceanography, University of California, San Diego, La Jolla, CA, United States of America
[7] NIOZ Royal Netherlands Institute for Sea Research and Utrecht University, Texel, Netherlands
[8] Smithsonian National Museum of Natural History, Washington, DC, United States of America
[9] CARMABI Foundation, Willemstad, Curaçao
[10] Department of Freshwater and Marine Ecology/Institute for Biodiversity and Ecosystem Dynamics, University of Amsterdam, Amsterdam, Netherlands
[11] Hawai'i Institute of Marine Biology, University of Hawai'i at Mānoa, Kāne'ohe, HI, United States of America
[12] Department of Biology, University of Miami, Coral Gables, FL, United States of America
[13] Organismic and Evolutionary Biology, Harvard University, Cambridge, MA, United States of America
* These authors contributed equally to this work.

## ABSTRACT

Reef-building corals are ecosystem engineers that compete with other benthic organisms for space and resources. Corals harvest energy through their surface by photosynthesis and heterotrophic feeding, and they divert part of this energy to defend their outer colony perimeter against competitors. Here, we hypothesized that corals with a larger space-filling surface and smaller perimeters increase energy gain while reducing the exposure to competitors. This predicted an association between these two geometric properties of corals and the competitive outcome against other benthic organisms. To test the prediction, fifty coral colonies from the Caribbean island of Curaçao were rendered using digital 3D and 2D reconstructions. The surface areas, perimeters, box-counting dimensions (as a proxy of surface and perimeter space-filling), and other geometric properties were extracted and analyzed with respect to the percentage of the perimeter losing or winning against competitors based on the coral tissue apparent growth or damage. The increase in surface space-filling dimension was the only significant single indicator of coral winning outcomes, but the combination of surface space-filling dimension with perimeter length increased the statistical prediction of coral competition outcomes. Corals with larger surface space-filling dimensions ($D_s > 2$) and smaller perimeters displayed more winning outcomes, confirming the initial hypothesis. We propose that the space-filling property of coral surfaces complemented with other proxies of coral competitiveness, such as life history traits, will provide a more accurate quantitative characterization of coral competition outcomes on coral

Corresponding author
Antoni Luque, aluque@sdsu.edu

reefs. This framework also applies to other organisms or ecological systems that rely on complex surfaces to obtain energy for competition.

## INTRODUCTION

Coral holobionts derive energy from photosynthesis—carried out by endosymbiotic algae—and heterotrophic feeding to build and maintain their calcium carbonate skeletons (*Porter, 1976*; *Okie, 2013*; *Madl & Witzany, 2014*). The construction of the skeleton leads to the overall structure of the coral colony, which yields specific geometric properties (e.g., surface area, perimeter, volume), and, thereby, shapes the geometry of coral reefs from cellular to ecosystem scales. Corals also compete with organisms for limited reef space (*Jackson, 1977*; *Meesters, Pauchli & Bak, 1997*), and these battles are fought along the border or perimeter of a coral colony at relatively small scales (µm–cm). Now, with modern 3D photogrammetry methods and 2D image merging software, the question of how coral morphology and its associated geometric properties relate to coral competition can be explored at high resolution (*Gracias & Santos-Victor, 2001*; *Burns et al., 2015*; *Leon et al., 2015*; *Young et al., 2017*; *Ferrari et al., 2017*; *Hatcher et al., 2020*; *Little et al., 2021*).

Competitive interactions generally occur along the coral's perimeter where polyps interact with benthic organisms such as fleshy algae, calcifying algae, sponges and other corals (*Jackson, 1977*; *Jackson, 1979*; *Buss & Jackson, 1979*; *Meesters, Wesseling & Bak, 1996*). At these interaction zones, a coral will either overgrow (win), be overgrown (lose), or display an apparent neutral interaction with the competitor species despite being in direct contact (Figs. 1A–1C) (*Jackson & Winston, 1982*; *Barott et al., 2012b*; *Swierts & Vermeij, 2016*). Loss of coral tissue also occurs due to disease and grazing (*De Bakker et al., 2016*; *Rempel, Bodwin & Ruttenberg, 2020*). Corals use several strategies to compete with benthic organisms including overgrowth (*McCook, Jompa & Diaz-Pulido, 2001*), the extension of sweeper tentacles and/or mesenterial filaments (*Chornesky & Williams, 1983*; *Nugues, Delvoye & Bak, 2004*; *Galtier d'Auriac et al., 2018*), and the use of microbial/chemical warfare (*Barott & Rohwer, 2012*; *Roach et al., 2017*; *Roach et al., 2020*). Therefore, a longer coral perimeter increases the number of competitive interactions which requires increased defenses compared to a shorter perimeter. The energy and resources required to defend a perimeter are obtained via photosynthesis and heterotrophic feeding across the entire surface area of a coral colony (*Porter, 1976*), and then distributed throughout the colony via the coenosarc tissue (*Rinkevich & Loya, 1983*; *Oren, Rinkevich & Loya, 1997*; *Henry & Hart, 2005*; *Schweinsberg et al., 2015*). As the surface area of a colony increases so does the number of polyps and the potential for energy acquisition and distribution (*Jackson, 1979*; *Oren et al., 2001*; *Okie, 2013*). Therefore, the relationship between perimeter length (i.e., energy needed for defense) and surface area (i.e., potential energy for defense and growth) may influence the result of the competition (e.g., win, lose or remain neutral).

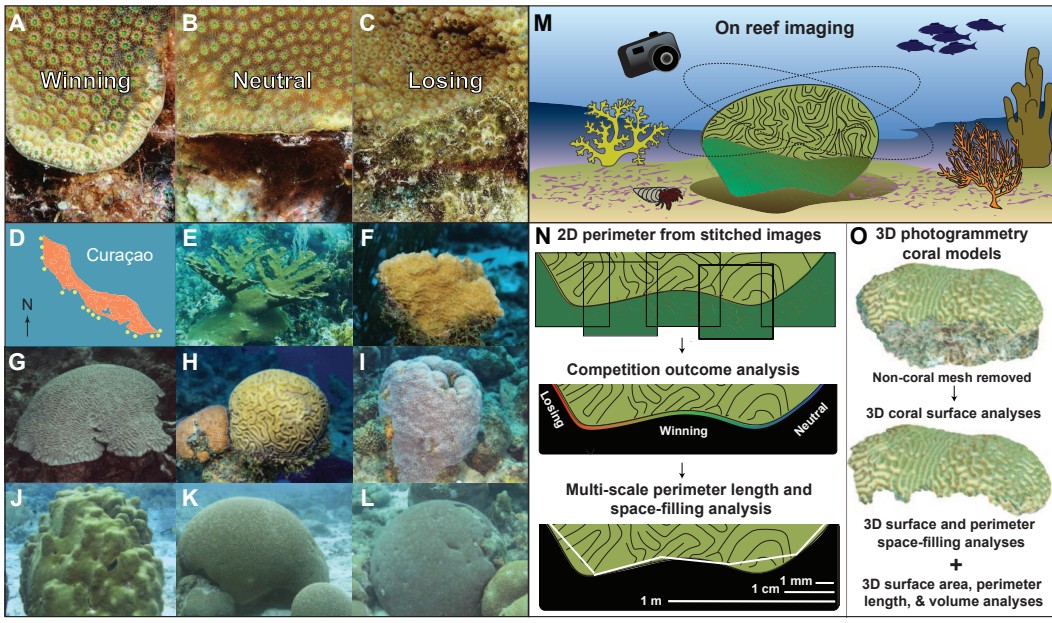

**Figure 1** **Coral competition outcomes and coral geometry methods.** The three possible outcomes of the coral competition are illustrated for the coral species *Orbicella faveolata*: (A) winning, the coral overgrows the algae; (B) neutral, neither coral nor algae is overgrowing one another; or (C) losing, algae are overgrowing the coral and coral skeleton is visible under the algae. (D) Curaçao site locations. The coral species studied include (E) *Acropora palmata*, (F) *Agaricia agaricites*, (G) *Colpophyllia natans*, (H) *Diploria labyrinthiformis*, (I) *Montastraea cavernosa*, (J) *Orbicella faveolata*, (K) *Pseudodiploria strigosa*, and (L) *Siderastrea siderea*. (M) Coral colonies were photographed in situ. (N) Close-range, overlapping pictures were stitched together to generate high-resolution 2D perimeter models. Outlines of the 2D coral perimeters were used to calculate the percentage of perimeter winning, losing and remaining neutral to competitors and to calculate the perimeter length and space-filling dimension over a 0.1 mm to 1 m scale range. (O) Photogrammetry methods were used to create 3D coral models, and non-coral mesh was removed to measure coral surface and perimeter space-filling dimension, surface area, perimeter length and volume.

Many coral geometric properties are dependent on coral morphology and life history. Massive corals with smaller perimeter-to-surface area ratios demonstrate greater resilience to algal overgrowth compared to encrusting corals with larger perimeter-to-surface area ratios (*Hughes, 1989*; *Tanner, 1995*; *Lirman, 2001*). Corals belonging to both morphological groups invest energy in defense and may be aggressive competitors against other benthic organisms (Fig. 1) (*Swierts & Vermeij, 2016*). In contrast, corals with branching and pillar-like morphologies have minimized their perimeters at the base of the colony and invest energy in vertical growth following the "escape in height" strategy (*Meesters, Wesseling & Bak, 1996*). Colony size has also been suggested to affect the outcome of coral-algal competition (*Sebens, 1982*; *Ferrari, Gonzalez-Rivero & Mumby, 2012*). However, some studies found that small and large corals were more effective at competing against algae than medium sized corals (40–80 cm) (*Barott et al., 2012a*), while others found that medium size corals were better competitors (*Swierts & Vermeij, 2016*). Additional studies are thus needed to determine the effect of coral morphology on coral competition outcomes.

A key factor that has been overlooked when assessing coral competition outcomes is the variation in the coral morphology and geometry across multiple structural levels (*Zawada, Dornelas & Madin, 2019*). Coral colonies with vastly different morphologies can display similar geometric properties, such as surface area or perimeter length, based on how corals fill a specific space. For example, a small branching coral may have the same surface area as a large mounding coral due to the space-filling nature of the branching coral. Maximizing surface area within a defined space is common in biological systems that seek to increase energy transfer or biochemical activity such as the inner mitochondrial membrane (*Faitg et al., 2020*) and the intestinal villi in animals (*Helander & Fändriks, 2014*). The opposite effect is expected for the coral perimeter where a decrease in space-filling would decrease the number of competitive interactions. Coral geometric properties are also dependent on the scale they are measured: surface area and perimeter length measured at the polyp-scale (mm) may be larger than at the colony-scale (m) due to the small-scale polyp features (*Holmes, 2008*). Therefore, how much space a coral surface and perimeter fills at a defined scale will have important biological implications.

The space-filling properties of surfaces and perimeters are assessed using fractal dimension metrics, which determine how the volume, surface, and perimeter of an object fill a space across multiple scales (*Mandelbrot, 1983*; *Murray, 2002*; *Falconer, 2003*; *Reichert et al., 2017*). Fractal dimensions have been used to analyze the geometry of natural systems from clouds and forests to plants, corals, and human organs (*Mandelbrot, 1983*; *Basillais, 1997*; *Reichert et al., 2017*). A physical property of an object—such as its volume, surface, or perimeter—is fractal if it satisfies two conditions (*Halley et al., 2004*). First, the fractal dimension of the measured property is different than its Euclidean dimension, that is, 2D for a surface or 1D for a perimeter. If the fractal dimension is larger than the Euclidean dimension, then the property increases exponentially at smaller scales, and if it is smaller, the property decreases exponentially as one magnifies the object (*Peitgen, Jürgens & Saupe, 1992*; *Cross et al., 1993*; *Falconer, 2003*; *Martin-Garin et al., 2007*). Second, the property must be self-similar or statistically similar across scales. The analysis of this second condition is more challenging, and many natural objects do not satisfy it (*Panico & Sterling, 1995*). Studies often use the term fractal when the first condition is met, even if the second condition is not confirmed or tested (*Murray, 2002*; *Halley et al., 2004*). To avoid abusing the term fractal and misleading interpretations, the term space-filling dimension will be used here instead of fractal dimension.

Space-filling studies on corals have analyzed the texture and complexity of coral structures at small and large scales (*Fukunaga et al., 2019*; *Zawada, Dornelas & Madin, 2019*; *Sous et al., 2020*). At the smallest scale, a box-counting method found that corallites (<10 mm) had a surface space-filling dimension of $D_s = 0.8$–$1.6$ which is below the Euclidean dimension of a surface ($D = 2$); this small space-filing dimension was associated with the cavities of the corallite, and the variance in space-filling dimension captured differences in corallite structures (*Martin-Garin et al., 2007*). At the coral colony level, cube-counting methods on 3D coral models found individual colonies to have space-filling dimensions up to 2.30, significantly larger than the Euclidean dimension of a surface ($D_s = 2$) (*Fukunaga & Burns, 2020*). At the reef level (~1 km), a variation method found

that the surface space-filling dimension was $D_s = 2.28–2.61$; this large value was associated to the rich texture of the reef (*Zawada & Brock, 2009*). The different levels of complexity is consistent with the space-filling dimension of the 2D perimeter ($D_p$) analyzed for large coral colonies (10–100 m, $D_P \sim 1.2$) and coral reefs (1 km, $D_P \sim 1.6$), both displaying larger values than the Euclidean dimension of a curve, $D = 1$ (*Bradbury & Reichelt, 1983*; *Mark, 1984*; *Purkis, Riegl & Dodge, 2006*). However, space-filling dimension studies on coral surfaces and perimeters at the colony level (millimeter to meter scales) have not been examined in the context of coral competition.

This study explored how Euclidean geometric properties and space-filling dimension of coral colonies were associated with coral competition outcomes (Figs. 1A–1C). We hypothesized that corals with larger surface space-filling properties and shorter perimeters would have more winning interactions than corals with smaller surface space-filling dimensions and longer perimeters. To test this hypothesis, the space-filling dimensions and Euclidean geometric properties of 50 coral colonies from the Caribbean island of Curaçao were obtained using 3D photogrammetry (Figs. 1D and 1M) and analyzed using simple and multiple regression methods. The results confirmed the hypothesis and showed significant relationships between coral geometric properties, including surface space-filling dimension and the competition outcome against benthic organisms.

## MATERIALS & METHODS

### Field sampling

Fifty coral colonies were randomly sampled from fifteen sites on the island of Curaçao (Fig. 1D) using the CARMABI field permit (2012/48584). This included four sites in the eastern (9 colonies), four sites in the central (37 colonies), and three sites in the western (4 colonies) regions of the island. The samples from the central region were more numerous due to favorable diving conditions, and corals were selected based on observations in the field. The coral species included *Acropora palmata* ($n = 2$), *Agaricia agaricites* ($n = 2$), *Colpophyllia natans* ($n = 5$), *Diploria labyrinthiformis* ($n = 4$), *Pseudodiploria strigosa* ($n = 8$), *Montastraea cavernosa* ($n = 10$), *Orbicella faveolata* ($n = 12$), and *Siderastrea siderea* ($n = 8$) (Figs. 1E–1L). The samples were skewed toward massive/boulder morphologies, reflecting coral species common in Curaçao (*De Bakker et al., 2016*), and the number of colonies per species reflected their relative abundance in the field. The depth ranged from 3.5 to 19 m, but the sampling was not designed to capture stratification. The 50 coral colonies were selected to reflect a range of sizes and interactions including fleshy algae, calcifying algae, sponges, other corals and/or no competitors (overhangs and sediment). The 50 colonies were photographed by SCUBA diving using a Canon Rebel T4i with a 35-mm lens and two Keldan 800 lumen video lights to illuminate the corals uniformly. An in-reef ruler was photographed with the corals to set the scale for the digital models, placing the ruler along the perimeter of the coral colonies.

### 2D perimeter models and competition outcomes

The 2D perimeter models were generated by photographing sections of the coral perimeter at close-range (Figs. 1A–1C). Multiple, overlapping pictures of the entire perimeter were
taken as close to the perimeter as possible (Fig. 1M). The high-resolution (mm) overlapping images of coral perimeters were stitched together to build each coral's 2D perimeter model (Fig. 1N) using Globalmatch and Guimosrenderer software, which was developed at Scripps Institution of Oceanography using methods from *Gracias & Santos-Victor (2000)*, *Gracias & Santos-Victor (2001)*; the software was accessed with permission through the Sandin Lab, Scripps Institution of Oceanography. All zones of the perimeter were considered an interaction zone, even the overhangs and sediment interactions, because corals growing on or near sediment were interacting with sediment-dwelling organisms (e.g., anemones) and fleshy algae and/or sponges were present on the underside of overhanging corals. Winning interactions were determined by coral tissue overgrowing or damaging the competitor, losing interactions were designated by apparent coral tissue damage or overgrowth by the competitor, and neutral interactions were identified if neither the competitor nor the coral showed signs of damage or overgrowth (Figs. 1A–1C). The interaction between corals and other competing organisms was outlined using a one-pixel pen tool in Adobe® Photoshop® CC 2014 at the edge of the coral tissue (Fig. 1N). The one-pixel wide outline was colored in separate RGB channels to designate competition outcomes: red (coral losing), green (coral winning), and blue (neutral). The fraction of red, green, and blue pixels was used, respectively, to obtain the percentage of losing (%L), winning (%W), and neutral (%N) interactions around a coral perimeter.

## 3D coral models and coral space-filling dimensions

Structure-from-Motion (SfM) photogrammetry was used to create 3D coral models. Autodesk® ReMake®, 2016 was utilized for 3D construction of coral models (*Burns et al., 2015*; *Leon et al., 2015*) (Fig. 1O) and measurement of geometric properties of corals such as perimeter, surface area, and volume (*Naumann et al., 2009*; *Lavy et al., 2015*). The initial coral renderings were investigated visually to identify patches of dead coral in the surface (Fig. S1) and patches without coral tissue were removed from the 3D models. The resolution of the final 3D models ranged from $1.4 \times 10^{-4}$ mm$^2$ to 0.75 mm$^2$ with an average of 0.070 mm$^2$, given as the median area of the triangles in the mesh of the 3D models.

Coral space-filling dimension was calculated using a box counting method (*Falconer, 2003*). The algorithm developed was generalized to multidimensional objects. The boxes corresponded to rectangles in 2D images and parallelepipeds in 3D images, and the initial box corresponded to the smallest axis-aligned bounding box containing the point cloud for each coral model. The logarithm of the number of boxes was plotted against the logarithm of the box size, and the space-filling dimension $D$ was extracted from the slope of the linear regression (Eq. S1, Fig. S2). The algorithm was validated against known fractal objects (*Peitgen, Jürgens & Saupe, 1992*; *Cross et al., 1993*; *Falconer, 2003*). These fractals were generated using seven recursion levels, and the box-counting used at least seven bisections (Table S1). The algorithm displayed an error smaller than 3%, and this error value was used as an upper theoretical error for the estimated space-filling dimension. The perimeter space-filling dimension ($D_P$) was calculated from the 2D models by applying the generalized box-counting algorithm using rectangular boxes. The 2D models were used due to their higher resolution with respect to the 3D model. The 2D models allowed

a minimum of ten bisections in the algorithm (smaller than 1 mm resolution), that is, a space-filling analysis across scales encompassing three orders of magnitude ($2^{10} = 1{,}024$). The surface space-filling dimension ($D_S$) was calculated from the 3D models by applying the generalized box-counting algorithm using parallelepiped boxes. A minimum of five bisections in the algorithm was used. Nonparametric bootstrap resampling was used to construct 95% bias-corrected and accelerated (BCa) confidence intervals on the slope ($D$) of the coral models (*Efron & Tibshirani, 1994*). Self-similarity over multiple scales was not tested and therefore, the fractality of the coral surface and perimeter was not determined. See Supplementary Material for additional details.

## Coral geometric properties: perimeter, surface area, volume, polyp size, and patch area

Coral perimeters were measured from the high-resolution 2D reconstructions as well as the 3D models. Both measurements were used because they had complementary advantages and disadvantages. The 2D reconstructions displayed higher-resolution at finer scales but distorted the overall shape of the perimeter due to the reconstructions. The 3D models captured the perimeter shape more accurately but the resolution at finer scales was lower, limiting the multiscale analysis. The perimeter length in the 2D reconstructions were obtained from the Richardson algorithm with a ruler size of 1 mm (*Mandelbrot, 1982*; *Falconer, 2003*). For volume calculations, the fill tool in Autodesk® Remake® was used to create a mesh that closed the area outlined by the perimeter of the 3D coral model, and then the volume of the 3D model was measured. Patches of dead tissue within the coral colonies were removed from 18 coral models to calculate total coral surface area and dead tissue patch area. Perimeter, surface area, and volume of the 3D models were calculated with the mesh report tool in Autodesk® Remake®, 2016. Polyp diameters were measured from the high-resolution perimeter images using ImageJ 1.47v, and 10 polyp diameters per colony were averaged. Additional details are provided in Supplementary Material, and the measurements for each variable are available in Data S1.

## Correlation with single variables

A least-squares linear regression was used to compare the percentages of losing (%L) and winning (%W) perimeter with respect to thirteen coral variables: depth (d), polyp diameter ($P_d$), volume (V), surface area (SA), volume-to-surface area (V/SA), surface area-to-polyp area ratio ($SA_{polyp}$), perimeter space-filing dimension ($D_P$), surface space-filling dimension ($D_S$), 2D perimeter length obtained from Richardson's algorithm ($P_R$), perimeter length obtained from 3D models ($P_{3D}$), 3D perimeter-to-polyp size ratio ($P_{polyp}$), 2D perimeter-to-surface area ratio ($P_R/SA$), and 3D perimeter-to-surface area ratio ($P_{3D}/SA$). These variables included absolute values of each coral colony as well as relative values with respect to polyp size since the energy and nutrient harvesting in the colony is obtained through polyps. The neutral interactions were a small fraction and were not studied in detail (Fig. 2). Nonparametric bootstrap resampling was used to construct 95% bias-corrected and accelerated (BCa) confidence intervals to validate the statistical significance of the slope *p*-values (*Efron & Tibshirani, 1994*). The outputs of the statistical analysis are available in Data S2.

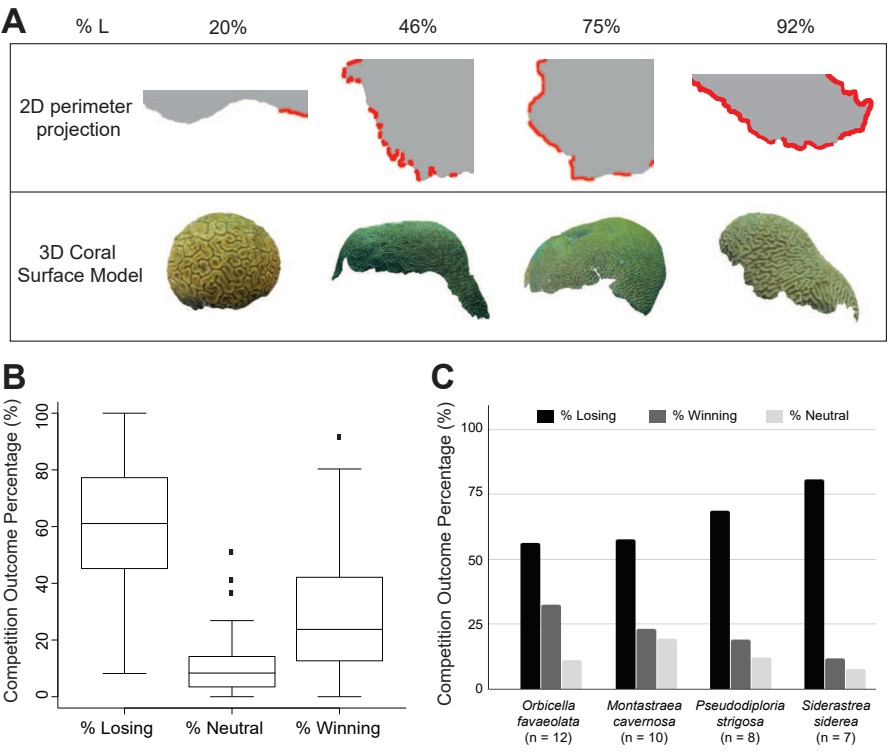

**Figure 2  Coral models and statistics for competitive outcomes.** (A) 2D coral perimeter models and 3D coral models with different percentages of losing perimeter (%L). Losing regions along the 2D perimeter projections are highlighted in red. (B) Box plots of the three perimeter outcomes: losing (%L), neutral (%N), and winning (%W) (Table S2). The middle line corresponds to the median, the range of the box contains the 25th to the 75th percentile, and each whisker is the minimum (in absolute value) between the 150% interquartile range (IQR) and the value of the most extreme point on that side of median. The black dots represent outliers exceeding the whiskers. (C) Bar chart of competitive outcomes for species with $n \geq$ 5. Bars show the average percentage of losing perimeter (black), winning perimeter (dark grey), and neutral perimeter (light grey).

## Multivariate regression using random forests

A multivariate regression analysis of the competitive outcomes as a function of the measured variables was performed applying the statistical learning method random forests using the R package randomForest (*Liaw & Wiener, 2002*). Random forests is a non-parametric approach, that is, it does not assume linear or non-linear relationships between variables, and the existence of correlations between input variables does not impact the output of random forests. Such variables are implicitly grouped with one dominating variable and the other variables as surrogates (*James et al., 2000*). Two independent approaches were used with random forests. First, all measured variables were incorporated in the initial model. The replication of the analysis using 100,000 random forests trials, however, indicated that the selection of significant variables and percentage explained variance was not robust. This was due to relatively small dataset (50 independent coral competitive outcome measurements) compared to the number of variables (14 in total: the 13 variables used in the univariate analysis plus the categorical species variable). Second, a bottom-up

alternative approach was followed. The statistical model for each response (%L and %W) was constructed by first selecting the variable with the highest univariate correlation and creating a random forests (RF) model. The remaining variables were added to the model using the following procedure. A variable was chosen from the unselected variables and added to the RF model, and the percent explained variance for this model was calculated. The variable was then removed from the model, and the next variable was added. This was repeated for all unselected variables, and the variable with the largest percent explained variance was marked as selected. The model was reset, and the entire procedure was repeated until 10,000 iterations were completed. The likelihood of variable selection was calculated as (# times selected)/(10,000). The variable with the highest likelihood of selection was added to the model, and the total number of variables in the model was increased. This procedure was repeated until the RF model contained all variables. Codes for all analyses are available on GitHub (https://github.com/luquelab/George_Mullinix_etal_2021).

## RESULTS

### Coral competition outcomes

The percentage of losing perimeter of sampled coral colonies ranged from 8% (*A. palmata*$_{CUR29}$) to 100% (*S. siderea*$_{CUR20}$). Figure 2A provides examples of coral 2D perimeters and 3D models displaying small (20%) to large (92%) percentage of losing perimeter. Corals displayed an average of 60% losing, 29% winning, and 11% neutral interactions along the perimeter (Fig. 2B and Table S2). Among species sampled in five or more colonies, *S. siderea* displayed the largest percentage of losing perimeter (81%), followed by *P. strigosa* (69%), *M. cavernosa* (58%), and *O. faveolata* (56%) (Fig. 2C). An inverse trend was found regarding the percentage of winning perimeter: *O. faveolata* (33%), *M. cavernosa* (23%), *P. strigosa* (19%), and *S. siderea* (12%). The percentage of neutral perimeter was considerably smaller and followed a different trend: *M. cavernosa* (19%), *P. strigosa* (12%), *O. faveolata* (11%), and *S. siderea* (8%). Overall, corals were losing a major fraction of their perimeters, and the neutral regions represented the smallest fraction among the three competitive outcomes. On average, *S. siderea* was the most vulnerable species, while *O. faveolata* was the most successful competitor.

### Space-filling dimensions of the coral perimeter and coral surface

The space-filling dimension, defined as the box-counting dimension, was interpreted with respect to the Euclidean dimension (ED) of the property measured, that is, $ED_S = 2$ for the surface and $ED_P = 1$ for the perimeter. If the surface space-filling dimension ($D_S$) of a coral was equivalent to the Euclidean dimension ($ED_S = 2$), then the surface would not strongly depend on the resolution of measurement, and the coral would be studied as a classical Euclidean object. If a coral instead had a surface space-filling dimension larger than two ($D_S > 2$), then the surface would be more complex, displaying additional features at higher resolutions. In this case, the total surface of the coral measured at 1 mm scale would be larger than at 1 m scale. Finally, if a coral had a surface space-filling dimension smaller than two ($D_S < 2$), it would lose features as the resolution increased due to the presence of patches devoid of coral polyps (Fig. S1), and the total surface area
would be smaller at higher resolutions. This concept also applies to the perimeter where the Euclidean dimension is $ED_P = 1$.

The perimeter space-filling dimensions ($D_P$) measured from the 2D perimeter reconstructions of the 50 corals were close to the Euclidean value $D_P \sim 1$, which was contained within the 95% confidence interval for all corals except for three colonies (Fig. 3A). Incorporating the theoretical error from the space-filling algorithm (3%) placed the three colonies within the Euclidean value: *O. faveolata*$_{CUR34}$ ($D_P = 1.00 \pm 0.03$), *S. siderea*$_{CUR54}$, ($D_P = 0.99 \pm 0.03$), and *A. palmata*$_{CSA142}$ ($D_P = 0.99 \pm 0.03$). The average space-filling dimension combining all corals was $<D_P> = 1.00 \pm 0.03$ (SE), and the mean values of individual corals ranged from 0.97 ($-2.65\%$) to 1.01 ($+1.17\%$) with respect to the average perimeter space-filling dimension (1.00). Coral perimeters with high ($D_P = 1.01 \pm 0.03$), medium ($D_P = 1.00 \pm 0.01$), and low ($D_P = 0.99 \pm 0.01$) space-filling dimension did not display any different salient geometric features (Fig. 3B). Thus, the apparently convoluted perimeters of coral colonies had a space-filling dimension consistent with the Euclidean dimension, and small differences in the space-filling dimension were not geometrically relevant.

The surface space-filling dimensions ($D_S$) for the 50 coral colonies were also close to the Euclidean value $D_S \sim 2$, which was contained within the 95% confidence interval for all corals except four: *O. faveolata*$_{CSA017}$ (1.94 $-$1.98 CI), *O. faveolata*$_{CUR34}$ (1.94 $-$1.95 CI), *M. cavernosa*$_{CUR40\_2}$ (1.90 $-$1.94 CI), and *O. faveolata*$_{CUR9}$ (1.84 $-$1.88 CI). When considering the error of the space-filling algorithm ($\sim$3%), *O. faveolata*$_{CSA017}$ ($D_S = 1.94 \pm 0.06$) and *O. faveolata*$_{CUR34}$ ($D_S = 1.94 \pm 0.06$) were statistically compatible with the Euclidean value, while *M. cavernosa*$_{CUR40\_2}$ ($D_S = 1.90 \pm 0.06$) and *O. faveolata*$_{CUR9}$ ($D_S = 1.86 \pm 0.06$) remained lower. The average space-filling dimension combining all corals was $<D_s> = 2.00 \pm 0.06$. The mean value of individual corals ranged from 1.84 ($-8.01\%$) to 2.13 ($+6.32\%$) with respect the average surface space-filling dimension. Corals colonies displayed significant geometrical differences as illustrated in Fig. 3C for corals with high (2.08 $\pm$ 0.04), medium (2.01 $\pm$ 0.04), and low (1.90 $\pm$ 0.03) surface space-filling dimensions. Corals with high surface space-filling dimensions had little to no patches of missing coral tissue and displayed a more texturized surface. Coral colonies with low space-filling dimensions instead displayed patches of missing coral tissue (Fig. S1), peninsula-like perimeters, and/or smoother/flatter surfaces (Fig. 3). Thus, the surface of coral colonies had space-filling dimensions near the Euclidean value, but they displayed a much larger relative variance with respect to the perimeter space-filling dimension and several contained salient geometric features between low and high surface space-filling dimensions.

## Relationship between competitive outcomes and individual geometric variables

The percentage of losing (%L) and winning (%W) perimeter was studied as a function of the 14 geometric and biological variables using linear regression analysis (Data S2). The surface space-filling dimension was the only variable that displayed a strong significant correlation with %L or %W (Figs. 4A and 4B). Percentage of losing perimeter (%L) correlated

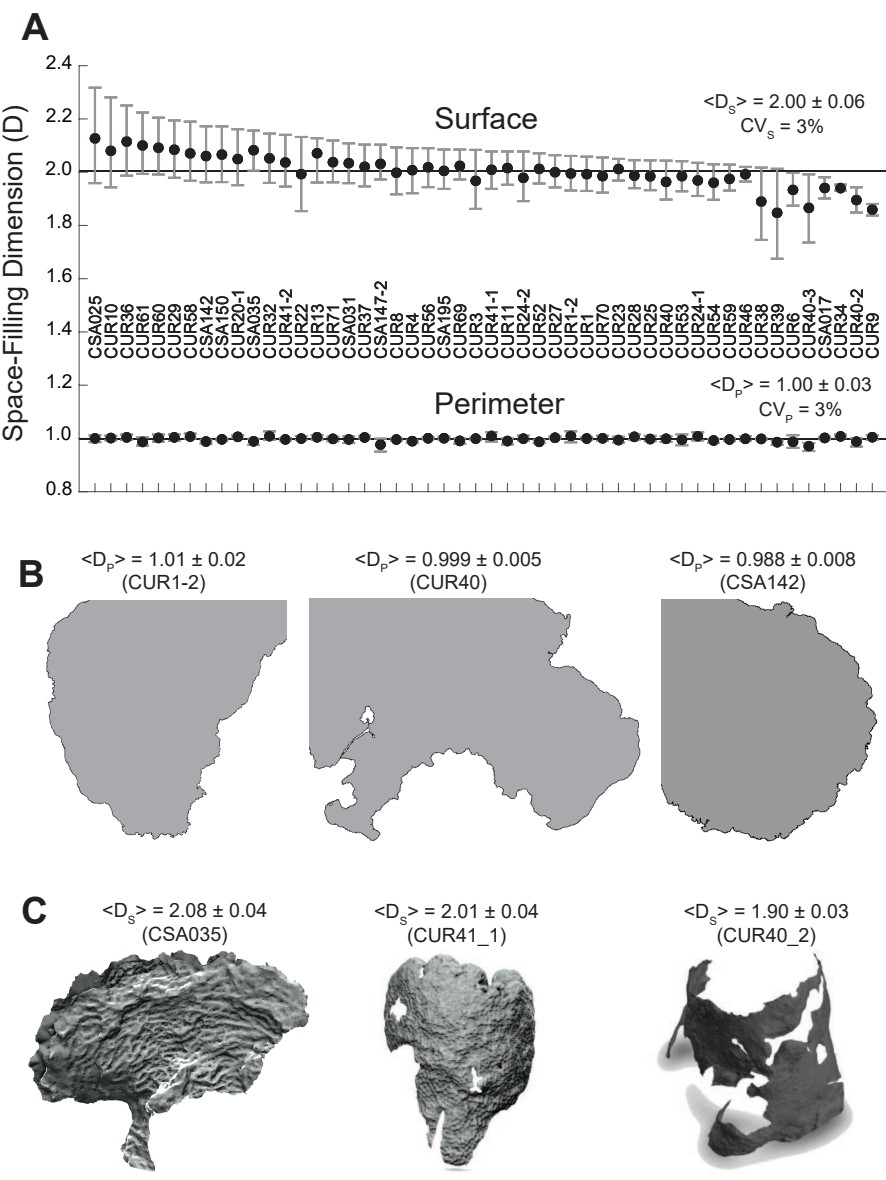

**Figure 3 Coral surface and perimeter space-filling dimensions.** (A) Plot of coral surface (top) and perimeter (bottom) space-filling dimensions with the means (black dots), 5–95% confidence intervals (whiskers), and labels associated with each coral sample. The solid line provides a reference for the Euclidean dimensions: $ED_p = 1$ (perimeter) and $ED_s = 2$ (surface). The mean values for the space-filling dimension of the perimeter ($D_p$) and the surface ($D_s$) ($\pm$ standard deviation) and their respective coefficients of variation (CV = standard deviation/mean * 100) are also included. (B) Two-dimensional coral perimeter models displaying space-filling dimensions below the Euclidean dimension. (C) Three-dimensional coral models associated with high, medium, and low space-filling dimensions for the coral surface. Empty spaces or holes in the 3D models show loss of coral tissue where the polyps have died or are overgrown by competing organisms.

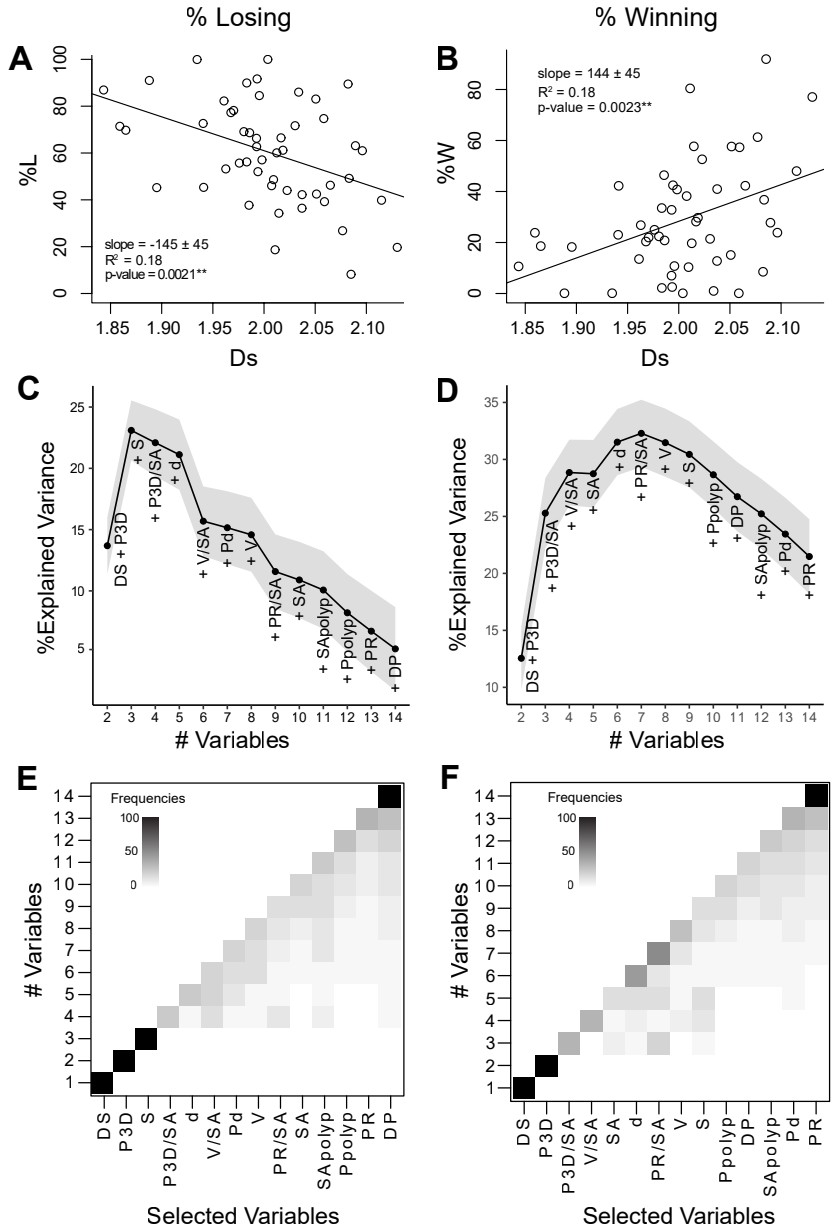

**Figure 4** **Relationships between coral geometry and coral competition outcomes: univariate and multivariate analyses.** The left panel of graphs relate to the percentage of losing perimeter (%L) and the right panel of graphs relate to the percentage of winning perimeter (%W). (A) and (B) Coral competition outcome as a function of the surface space-filling dimension with a linear regression fitted using the least-squares method. Coral surface space-filling dimension is the best single indicator of coral competition outcomes. (C) and (D) Bottom-up random forest model construction of coral geometric variables with the explained variance plotted as each variable is added. The solid black line corresponds to the average variance explained (%), and the grey area represents the 95% confidence interval. (E) and (F) Variable selection likelihood maps where the frequency (%) of the selected variable is shown. Frequencies range from 0% (white) to 100% (black). Variables include surface space-filling dimension ($D_S$), perimeter from 3D models ($P_{3D}$), species (S), 3D perimeter-to-surface area ratio ($P_{3D}/SA$), depth (d), volume-to-surface area (V/SA), polyp diameter ($P_d$), volume (V), 2D perimeter-to-surface area ratio ($P_R/SA$), surface area (SA), surface area-to-polyp area ratio ($SA_{polyp}$), 3D perimeter-to-polyp size ratio ($P_{polyp}$), perimeter from 2D models ($P_R$), and perimeter space-filing dimension ($D_P$).

negatively with surface space-filling dimension (slope $= -145 \pm 45$, $R^2 = 0.18$, $p$-value $= 0.0021$**), while percentage of winning perimeter (%W) correlated positively with surface space-filling dimension (slope $= 144 \pm 45$, $R^2 = 0.18$, $p$-value $= 0.0023$**). This was consistent with %W being negatively correlated with %L (slope $= -0.9 \pm 0.1$, $R^2 = 0.8$, $p$-value $= 2.2 \times 10^{-16}$ ***) (Fig. S3). The surface area was the only other variable that came close to being significant, but it displayed a much weaker correlation with the competitive outcomes (%L: $R^2 = 0.09$, $p$-value $= 0.05$; %W: $R^2 = 0.08$, $p$-value $= 0.05$) (Fig. S4). Thus, the mean values of the surface space-filling dimension displayed the strongest correlation with competitive outcomes, capturing 18% of the variance (Figs. 4A and 4B, $R^2 = 0.18$).

Coral competitive outcomes and geometric variables were also analyzed separately for species represented by more than five sampled colonies: *Orbicella faveolata* ($n = 12$), *Montastraea cavernosa* ($n = 10$), *Pseudodiploria strigosa* ($n = 8$), and *Siderastrea siderea* ($n = 7$). All species displayed a decrease in the percentage of losing perimeter (%L) as surface space-filling dimension increased (Fig. S5). However, the relationship was statistically significant only for *S. siderea*, and the variance in the other species was too large with respect to the sample size. No other variables significantly correlated with %L or %W in the species-specific analysis.

## Importance of combined geometric variables in coral competition outcomes

The bottom-up random forests (RF) model approach yielded a robust, repeatable construction of the non-parametric statistical model. For the percentage losing perimeter (%L), the addition of perimeter ($P_{3D}$) and species (S) to the surface space-filling dimension ($D_s$) sharply increased the percentage explained variance in the model to $23 \pm 1\%$ (Fig. 4C). Adding more variables, however, decreased the percentage of variance explained in the model due to the limited dataset size ($n = 50$). The selection of perimeter ($P_{3D}$) and species (S) as the second and third most important variables, respectively, was very robust with a frequency close to 100% (Fig. 4E). For the percentage winning perimeter (%W), the addition of perimeter ($P_{3D}$) and perimeter-to-surface area ($P_{3D}$/SA) to the surface space-filling dimension ($D_s$) sharply increased the percentage explained variance in the model to $25 \pm 2\%$ (Fig. 4D). The addition of variables involving coral volume-to-surface area (V/SA), surface area (SA), sampling depth (d), and projected perimeter-to-surface area ($P_R$/SA) led to a maximum percentage explained variance of $32 \pm 1\%$. The addition of more variables decreased the percentage variance explained due to the limited size of the dataset ($n = 50$). The selection of the perimeter ($P_{3D}$) as the second most important variable had a frequency of 100% (Fig. 4F). The subsequent variables were not always selected in the same order of importance. However, in both analyses (%L and %W), the frequency of the least important variables selected were robust (Figs. 4E and 4F). In particular, the perimeter space-filling dimension was selected as the least relevant variable in predicting coral competitive outcomes. This is consistent with the fact that no apparent geometric features were observed in the initial space-filling analysis of the perimeter (Fig. 3). The results for the percentage losing and winning perimeters therefore indicated that

the surface space-filling dimension and perimeter length were the most important variables associated to the coral competition outcomes.

## Hierarchical analysis of coral competitive outcomes and coral geometry

When analyzing the hierarchy of the most relevant geometric variables for the percent of losing perimeter (%L) and winning perimeter (%W), the primary node corresponded to the surface space-filling dimension, and the secondary node was the perimeter from 3D models (Fig. 5). For the percent losing tree (Fig. 5A), corals with $D_S \geq 2$ had the smallest %L, and among those, corals with $P_{3D} < 91$ cm formed the group with the smallest percentage of losing perimeter (24%). A similar trend was found for the percent winning tree (Fig. 5B): corals with $D_S \geq 2$ and $P_{3D} < 91$ cm had largest percentage of winning perimeter (84%). For corals with $D_S < 2$, smaller perimeters related to increased %L and decreased %W compared to corals with $D_S < 2$ and larger perimeters. The fact that small perimeters can have opposite effects depending on the surface space-filling dimension may explain why the perimeter length did not display a correlation in the single variable analysis. Overall, these results suggest that corals with a surface space-filling dimension $D_S \geq 2$ and relatively small perimeters have better competition outcomes (i.e., less losing and more winning outcomes).

# DISCUSSION

## Hypothesis: surface space-filling dimension and coral competitive edge

This study hypothesized that an increase in surface space-filling dimension and a reduction in perimeter would be associated with successful coral competitive outcomes. The rational was that the complexity added by the surface space-filling dimension would increase the potential for energy harvesting without necessarily leading to a larger coral colony and exposing more perimeter to benthic competitors. The univariate, statistical analysis indicated that the surface space-filling dimension was the most important variable associated to the percentage of winning and losing perimeter, and the multivariate analysis confirmed that the perimeter was the second most important variable (Fig. 4). The selection of these variables and their associated trends with coral competitive outcomes supported the initial hypothesis (Fig. 5). Therefore, this study indicates a potential causality of higher surface space-filling dimensions in the resilience and health of corals, and longitudinal studies that assess changes in competitive outcomes over time will be necessary to confirm this prediction.

## Relationship between coral geometry and coral interaction outcomes

The optimal combination of coral geometric properties explained $23 \pm 2\%$ to $32 \pm 1\%$ of the coral interaction outcomes, where the surface space-filling dimension was the best single indicator for the percentage of losing or winning perimeter (Fig. 4). The space-filling dimension or complexity of a coral surface likely reflects the ability to harvest energy through heterotrophic feeding of individual polyps and photosynthesis carried out by the

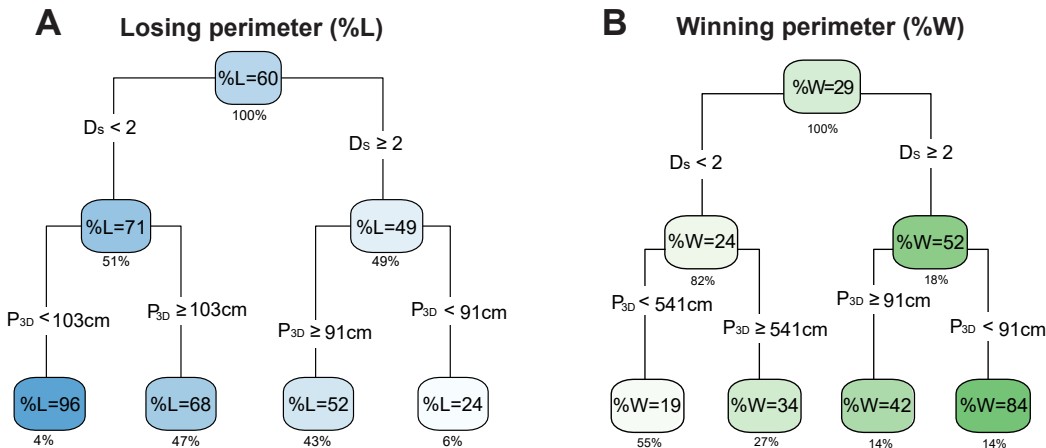

**Figure 5** **Interdependence of optimal variables in the predictions of outcomes.** Average regression trees generated for the percentage of (A) losing perimeter (%L) and (B) winning perimeter (%W) including the selected variables in the refined random forests analysis (Fig. 4). Each cluster displays the average outcome and the value below the box indicates the percentage of data contained in the cluster. The intensity of blue is proportional to %L and the intensity of green is proportional to %W. Variables include the perimeter length from 3D models ($P_{3D}$) and the coral surface space-filling dimension ($D_s$).

endosymbiotic *Symbiodiniaceae* (*Bachar et al., 2007*). Losing corals had surface space-filling dimensions below the Euclidean dimension ($D_S < 2$), and some of the reduced complexity was associated with patches (absence of coral tissue) and small, smooth colony surfaces (Fig. 3C, Fig. S1). The patches of missing coral tissue reduce the network of polyps, thus impeding the allocation of resources. However, patch area alone failed to correlate with either competitive outcomes or space-filling dimension (Fig. S6), and significant trends between competitive outcomes and space-filling dimension were also observed in corals without patches (Fig. S7). Therefore, the patches of dead tissue are only one of multiple features relevant to coral competitive outcomes measured by the space-filling dimension metric.

The perimeters of coral colonies with low space-filling dimensions also displayed large peninsula-like shapes that increase the perimeter length along with the potential number of competitive interactions (Fig. S1). The two features, dead tissue patches and perimeter peninsulas, may elucidate the low surface space-filling dimension of corals with larger percentages of dead or overgrown tissue along the perimeter. Conversely, winning corals had surface space-filling dimensions larger than the Euclidean dimension ($D_S > 2$) which translated into large surface areas with more complex surface features and no patches (Fig. 3C). This implies a denser network of polyps to feed and harvest energy, thus explaining the larger percentage of positive competitive outcomes for coral colonies with larger surface space-filling dimensions.

In addition to surface space-filling dimension, the perimeter length was another significant geometric variable in coral competition outcomes (Fig. 4). Corals with larger surface space-filling dimensions and smaller perimeters displayed increased %W and decreased %L (Fig. 5). This supports the hypothesis that more energy is required to defend

a longer perimeter, and that corals with complex surfaces may provide additional resources for defense compared to corals with less complex surfaces. However, smaller perimeters appeared unfavorable to corals with surface space-filling dimensions smaller than two (Fig. 5). This is a surprising result, but the corals in these categories had the highest %L (96%) and lowest %W (19%). The corals also displayed relatively smooth perimeters with no peninsula-like features compared to corals with $D_S < 2$ and longer perimeters. The loss of perimeter features potentially correspond to significant coral tissue damage around the perimeter, and despite the reduced perimeters, the corals may have lost too much surface complexity, thus reaching an unavoidable trajectory to be overgrown by competitors. However, the dynamics of coral competitive outcomes need further investigation to determine how these coral geometrical properties relate to the resilience of corals over time.

Surface area had significant correlations with competition outcomes (Fig. S4), but the relationship was weaker than the surface space-filling dimension. This suggests that the increase in surface area per $cm^2$ captured by the space-filling dimension provides a more accurate account for coral competitiveness than surface area alone, and that a larger surface area also does not necessarily correspond to a higher space-filling dimension. For example, *Orbicella faveolata*$_{CUR36}$ displayed a higher space-filling dimension than *Orbicella faveolata* $_{CUR08}$ and also had a greater percentage of winning perimeter despite its smaller surface area (smaller colony). The information regarding the relative increase in effective surface area per $cm^2$ captured by the space-filling dimension justifies the higher explained variance obtained by this metric compared with surface area alone.

Species-specific life traits also played a role in competition outcomes since coral species was a relevant variable in the multivariate analysis for the percentage of losing perimeter (Fig. 4). Several of the species studied had similar mounding morphologies (Figs. 1G–1L), although they differ in photosynthesis rates, growth rates and defense strategies (*McCook, Jompa & Diaz-Pulido, 2001*; *Roth, 2014*). *Orbicella faveolata* has higher maximum photosynthesis rates, photochemical efficiency and symbiont density compared to other species such as *M. cavernosa*, *P. strigosa* and *S. siderea* (*Muthiga & Szmant, 1987*; *Warner, Fitt & Schmidt, 1996*; *Castillo et al., 2014*; *Scheufen, Iglesias-Prieto & Enríquez, 2017*). Out of these species, *O. faveolata* displayed the largest average surface space-filling dimension, suggesting that the space-filling property of *O. faveolata* is maximized for photosynthesis. The greater photosynthetic capabilities of *O. faveolata* may also explain the competitive outcome results, where *O. faveloata* had the greatest percentage of winning interactions compared to the other three species. The two species, *P. strigosa* and *S. siderea*, are considered weedy coral species, and they had the greatest percentage of losing perimeter (*Toth et al., 2019*). On the other hand, *Montastraea cavernosa* is a slow growing species (*Manzello et al., 2015*) that had few losing interactions, suggesting that slower growing corals invest more resources in protecting their perimeters, as observed in the increased sweeper tentacles near interaction zones of *M. cavernosa* colonies (*Chornesky & Williams, 1983*). The energy tradeoff between growth and defense directly affects the competition outcomes along the perimeter of a coral (*Swierts & Vermeij, 2016*), and different coral species have evolved various strategies to balance this tradeoff. Our study

focused on mounding morphologies (Figs. 1E–1L), and additional data for species with other morphologies and life traits will be necessary to confirm the relationship between geometric properties and coral competitive outcomes.

The combination of surface space-filling dimension, perimeter, and additional geometric properties were able to explain up to 30% of the variance in the coral competition outcomes. Methodologies that improve the 3D coral model reconstruction and increase the signal-to-noise ratio may lead to geometric analyses with higher explained variances. Additionally, other factors are known to be relevant in determining the outcomes of coral space competition with other sessile organisms. Physiological factors display strong codependences with the geometry (*Jackson, 1979*; *Merks et al., 2004*; *Zawada, Dornelas & Madin, 2019*), but another factor that may be orthogonal and complementary to the geometry is the assembly of microbes associated with the coral and its competitors. Microbes and their associated metabolites are important in mediating the ecological interaction of corals with other benthic organisms (*Haas et al., 2016*; *Silveira et al., 2019*; *Roach et al., 2020*), along with coral disease (*De Bakker et al., 2016*). The combination of geometry, species-specific life traits, and microbial properties will improve the prediction of coral competition outcomes in future studies.

## Coral space-filling dimensions over multiple scales

The space-filling dimensions of the coral colonies obtained from this study were also compared to prior space-filling or fractal studies encompassing smaller (coral septa or polyp) and larger (reef) scales (Fig. 6). The qualitative analysis indicated that the space-filling dimension of corals increased from millimeter to kilometer scales, where the space-filling dimensions were closer to the Euclidean dimensions at the colony scale investigated here. This is a potential consequence of different physical factors and biometric allometry across scales. Water moves via diffusion in small-scale (µm to mm) boundary layers close to coral surfaces, and moves by advection at larger scales (m) due to wave motion and currents passing over reefs (*Barott & Rohwer, 2012*). A change in mechanisms from small to large scales likely leads to scale-dependent evolution of calcium carbonate structural architecture that resists water movement (e.g., a single corallite compared to an entire coral reef). In fact, the space-filling dimensions at the coral reef scale displayed similar values to seagrass beds and hard ground patches (*Zawada & Brock, 2009*), suggesting that the topography of the underlying substrate is responsible for the increased space-filling dimension at these larger scales. However, these calculations were constrained by the models' resolution, and future methods that implement a continuum of mm to km scales will help identify changes in reef space-filling properties over spatial scales.

The surface space-filling analysis and coral competition outcomes also extend to the reef level where large-scale interactions between coral reefs, sand flats and algae-dominated patches occur. The methods presented here also provide a framework for the study of other marine or even terrestrial systems. For example, crustose coralline algae are important reef calcifying phototrophs that rely on surface area to obtain energy and compete with other benthic organisms including coral. These methods are also applicable to terrestrial systems like forests where trees acquire energy through photosynthesis and obtain nutrients

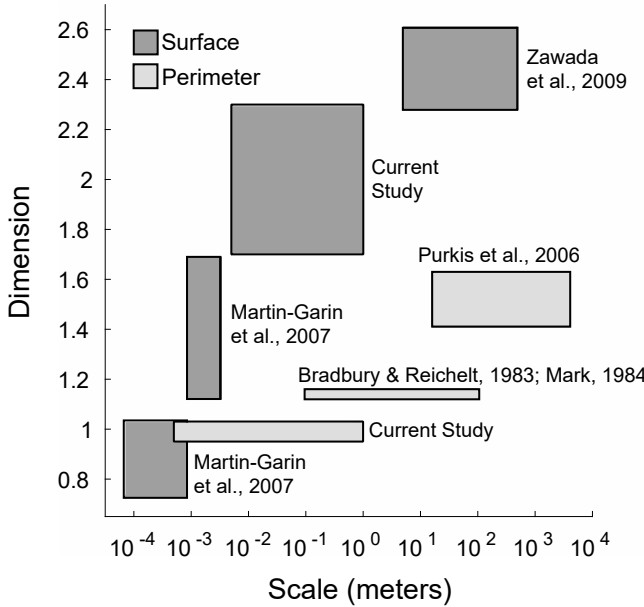

**Figure 6** **Coral space-filling dimension from corallite to reef scales.** The plot shows the range of space-filling dimensions measured across various scales from different coral studies. The space-filling dimensions are grouped in two categories: Surface space-filling dimension (dark grey) and perimeter space-filling dimension (light grey). For the perimeter, the ranges correspond to coral colonies (current study—box-counting method), larger coral colonies (*Bradbury & Reichelt, 1983*; *Mark, 1984*)—step-counting method), and coral reefs (*Purkis, Riegl & Dodge, 2006*—box-counting method). For the surface, the ranges correspond to corallite texture (*Martin-Garin et al., 2007*)—box-counting method), corallite structure (*Martin-Garin et al., 2007*)—box-counting method), coral colonies (current study—box-counting method), and coral reefs (*Zawada & Brock, 2009*—variation method).

through their roots, which display space-filling properties, while also shunting resources to other trees through mycorrhizal networks (*Simard et al., 1997*). Therefore, this work is pertinent to organisms or ecological systems that rely on surfaces for energy acquisition and share resources through biological networks.

## CONCLUSIONS

This study found a significant association between coral geometry and coral competition outcomes, explaining up to 32% of the variance in a diverse dataset. Corals with losing outcomes had low surface space-filling dimensions ($D_S < 2$) and displayed patches with no polyps and large peninsulas, while winning corals ($D_S > 2$) were more compact and displayed more complex and rugose surfaces with smaller perimeters. These findings support the hypothesis that larger surface space-filling dimensions favor energy harvesting, and smaller perimeters reduce the invasion of benthic competitors, providing a competitive edge to corals with these properties. This approach also provides a framework to study coral space-filling and benthic competitions at large reef scales along with the study of other organisms or ecological systems that rely on complex surfaces to obtain energy for competition.

## ACKNOWLEDGEMENTS

We thank Mark Hatay for the original artwork that was adapted to generate Fig. 1M.

### Funding

The work of Antoni Luque was funded by the National Science Foundation Award 1951678 in the Division of Mathematical Sciences. The work of Forest Rohwer and Aaron Hartmann was funded by the PIRE grant: NSF Partnerships for International Research and Education Grant (1243541). The work of James Mullinix was also supported by a STEM scholarship award funded by the National Science Foundation grant DUE-1259951. Additional funding was provided by the University of British Columbia International Doctoral Fellowship. The funders had no role in study design, data collection and analysis, decision to publish, or preparation of the manuscript.

### Grant Disclosures

The following grant information was disclosed by the authors:
National Science Foundation Award: 1951678.
PIRE.
NSF Partnerships for International Research and Education Grant: 1243541.
National Science Foundation: DUE-1259951.
University of British Columbia International Doctoral Fellowship.

### Competing Interests

The authors declare there are no competing interests.

### Author Contributions

- Emma E. George, James A. Mullinix and Antoni Luque conceived and designed the experiments, performed the experiments, analyzed the data, prepared figures and/or tables, authored or reviewed drafts of the paper, and approved the final draft.
- Fanwei Meng analyzed the data, prepared figures and/or tables, and approved the final draft.
- Barbara A. Bailey, Clinton Edwards, Ben Felts, Aaron C. Hartmann and Peter Salamon analyzed the data, authored or reviewed drafts of the paper, and approved the final draft.
- Andreas F. Haas, Benjamin Mueller, Mark J.A. Vermeij, Cynthia Silveira and Ty N.Y. Roach performed the experiments, authored or reviewed drafts of the paper, and approved the final draft.
- Forest Rohwer conceived and designed the experiments, performed the experiments, analyzed the data, authored or reviewed drafts of the paper, and approved the final draft.

### Field Study Permissions

The following information was supplied relating to field study approvals (i.e., approving body and any reference numbers):

Field experiments were conducted under the CARMABI research station permit (2012/48584).

## Data Availability

3D coral models and 2D perimeter models are available at Dryad: https://doi.org/10. 5061/dryad.5x69p8d2x.

Raw measurements along with the results from the univariate and Random Forest analyses are available in Supplementary Files.

Code for the CUDA-based Bootstrap application for the linear regression algorithm is available at GitHub: https://github.com/luquelab/George_Mullinix_etal_2021.

## Supplemental Information

Supplemental information for this article can be found online at http://dx.doi.org/10.7717/ peerj.11213#supplemental-information.

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
