# Peer review of "Space-filling and benthic competition on coral reefs"

_PeerJ, doi:10.7717/peerj.11213_

## Round 0.1 · original submission · Major Revisions

To provide a better understanding of this topic, in addition to recommendations/comments made by the reviewers, the authors shall consider how to make the ms easy for general readers to understand and also put everything in a better ecological context.

Reviewer 1 ·

Basic reporting

The paper is well written. The idea of structural complexity affecting competitive outcomes is very interesting. The references overall seem a bit old. I understand that 3D reconstruction of corals/coral reefs is still a new approach, but there has been a lot of research in the last few years, and some of them do look into fractal dimension calculated from coral colonies/reefs.

Here are a couple of specific comments related to background information.

L 99-101. "Coral geometry is generally measured at one scale using Euclidean shapes, but this method fails to capture the complexity of coral geometry."

What are examples of the coral geometry measured at one scale and how do they fail to capture the complexity? Please briefly explain.

L 111-128. This paragraph talks about fractal dimension values from different studies. There are different methods to obtain fractal dimension metrics, and they are not directly comparable (e.g. see Fukunaga, A.; Burns, J.H.R. Metrics of Coral Reef Structural Complexity Extracted from 3D Mesh Models and Digital Elevation Models. Remote Sens. 2020, 12, 2676.) Please clearly define what type(s) of fractal metric these studies that are cited in this paragraph used, and more importantly which specific fractal metric authors used in the present study.

Experimental design

Authors' research question is very well defined. More detailed description, especially of the calculation of fractal dimension, should improve the manuscript.

Here are some specific comments.
L 186-191. Regarding the calculation of surface fractal dimension, was this done in a true 3D space? Please describe more details. How did you determine the placement of boxes encasing the structure? Did you use any software or any custom scripts?

L 191-195. Is the perimeter fractal dimension still based on a box-counting method? Please clarify.

L 195-198. The surface fractal dimension was calculated from 3D models (so it was more like a cube-counting) and the perimeter fractal dimension was from 2D models using boxes, correct? I'm a bit confused. Also, was the perimeter fractal dimension obtained from the entire coral colony, not just "perimeter"? So, the 2D models used were basically models captured from the over-head angle? Or, was it just the "base" of each colony. i.e. If there is a branching morphology, was is the entire colony captured from the over-head angle or just a small round portion at the base of the colony (the substratum-coral interface)?

L 237. Did you check for multicolinearity? Or, is this particular statistical method robust to such an issue. Please clarify for those who are not familiar with the random forest method as there are some many variables in the model.

Validity of the findings

I am not familiar with the particular statistical method used in the study (i.e. random forest), so I cannot comment there.

Here are some specific comments about fractal dimension.

L 276-278. "Finally, if a coral had a surface fractal dimension smaller than two (DS < 2), it would lose features as the resolution increased due to the presence of holes devoid of coral polyps, and the total surface area would be smaller at higher resolutions."

Has this been proven using the specific method of fractal dimension calculation used in this study (if so, please cite the study)? Estimation of fractal dimension using the slope of regression line sometimes has problems, and I am not fully convinced that this explanation is true.

L 287-289. "Coral perimeters with high (DP = 1.01 ± 0.03), medium (DP = 0.999 ± 0.01), and low (DP = 0.988 ± 0.01) fractal dimension (±SE) did not display any different salient geometric features (Figure 3B)."

I am extremely puzzled by these values less than 1. Fractal dimension is a measure of irregularity, so fractal dimension less than 1 means that the perimeter was less irregular than a straight line. I know it happens sometime with some methods of fractal dimension calculation. Is this (i.e. fractal dimension being less than 1) common for this specific method of fractal dimension calculation used in the present study? The same goes to the surface fractal dimension being less than 2.

L 305-306. "Coral colonies with low fractal dimensions instead displayed holes, peninsula-like perimeters, and smoother surfaces."

Are these "holes" part of a natural coral morphology? It is hard to image "holes" in a coral colony. Where do they come from? If they are due to photo alignment issues, the models should not be used.

Additional comments

I find this study very interesting and the use of multiscale metric such as fractal dimension in a coral reef study is very exciting. My main concern with this paper is the lack of clarity in the calculation method of fractal dimension and references to the topological dimension (Dp = 1, Ds = 2). It is very difficult to accept that a surface and a perimeter are less complex than a horizontal plane and a straight line, and it makes me wonder if this is a some kind of artifact related to the calculation method. The paper also references to previously published values of fractal dimension without specifying what kind of fractal dimension was used in each study. Fractal dimension values obtained using different methods are not directly comparable.

I also wonder why authors did not try something a bit more simple, such as 3D surface complexity (the ratio of 3D surface area to 2D surface area, similar to the traditional rugosity measure in coral reef ecology). This would be much simpler to interpret as 3D surface complexity for sure means a horizontal plane, and it seems sufficient to answer authors question.

My suggestion is therefore either to add more detailed information on fractal dimension and provide some proof that Dp < 1 or Ds < 2 are not due to some sort of artifact or to use a metric other than fractal dimension, such as 3D surface complexity.

·

Basic reporting

Language is mostly good, but see my specific comment in the attached pdf.

Experimental design

see attached pdf

Validity of the findings

see attached pdf

Additional comments

My recommendation is between a minor and major revision. The research question is interesting and the approach is suitable. However, I think the paper is overcomplicated in parts. It's an ecological study, but the content is mostly mathematical and it lacks connection between the mathematical descriptions and what they mean in terms of coral biology. In order to make this study more interesting to ecologists / biologists it needs to be simplified a bit and placed in a more ecological context. For example, I'm sure there are studies out there that looked at metabolic rates of at least some of the coral species investigated. Does the assumption that certain morphologies are able to gain more energy per surface area (either through autotrophy or heterotrophy) than others hold true?

Reviewer 3 ·

Basic reporting

See general comments.

Experimental design

See general comments.

Validity of the findings

See general comments.

Additional comments

This study related competitive outcomes amongst corals to structural characteristics (derived from 3D and 2D photo-based modelling) of individual colonies. The work used 50 colonies from 8 different species at several sites around the island of Curacao. The study concluded that the surface fractal dimension was able to explain (positively) a significant amount of variation in competitive success amongst colonies and that the addition of the perimeter distance increased this variance explained (negative relationship to competitive success). The authors indicated this fits with their hypotheses that greater amounts of tissue available for acquisition of energy should aid in competition while longer perimeters require more resources to defend.

In general I really like this study. It’s uses some novel approaches to quantify 3D structure in corals to evaluate some fundamental ecological processes on reefs and does so by building on a host of previous studies. The study brought together nicely many different threads of knowledge about what mitigates competitive outcomes where corals fight for space. I think the work will have a wide appeal. I did have some comments/concerns which I outline below

Major comments
The major conclusion from the study is that surface fractal dimension (Ds) relates positively to successful outcomes in competition. The explanation for this is that high Ds values reflect an exponential increase in the amount of surface area at increasingly smaller scales. This is based on an accurate characterisation of what D fundamentally is. Now fractal dimension is a slippery concept and some would argue that having a D value greater than the fundamental dimension may not automatically indicate something is behaving as a fractal as its relatively easy to get a linear descending fit in a box-counting algorithm to any surface on a log-log scale. But leaving that argument aside, its worth stepping back and asking, if the pattern observed here is driven as suggested, by energy acquisition, would we not expect also to see a relationship just to surface area alone? Why is that not apparent and what does it meant that it is not? Related to this, I was also a bit worried about the effect of holes in the models. I believe these were areas that were trimmed out because they were dead. And much of the discussion about models with low Ds values centred around them being the ones with holes. And indeed, thinking about the box counting algorithm works, I can see where the effect of this would be to reduce the number of boxes with mesh in them increasingly as box size gets smaller, thus flattening the curve and giving a lower Ds. But could the explanation not also be a bit more direct in that corals with lots of dead tissue are fundamentally less healthy and thus are not doing as well in these competitive interactions at their margins? I would think this would need to be evaluated before a conclusion about Ds could be made so conclusively.

I was surprised not to see more analysis focused on the issue of species effects. The paper is written as though the effects here essentially supersede any from species alone. This would be a very interesting conclusion indeed. But I don’t think any of the models looked at interactive effects of the structural variables with species, which would be a necessary analysis to back this up.

I liked the approaches to looking at perimeter length but I think there was some further explanation needed. For instance, when this concept is introduced very early (ln 81) its not clear how one gets this when most of the discussion is about 3D coral colonies. What would perimeter mean in that context? Then when it is explained more fully, the acutal method used to stitch images together is not very clear. Perhaps this is because I was expecting perimeter to based on a 2D project of the 3D model and was preparing to hear some justification of which projection plane was used. So I think it would be worth mentioning at the start of the section (ln 160) that different approaches to perimeter were used as well as the merits of each to justify it.

As the study is all about explaining patterns of competition, it becomes important that the method for selecting colonies does not introduce any potential bias. Not much is said about the actual method for selecting colonies (but should be) other than that they were away from other corals. I was not clear why this was done, given the objective of looking at competition.

Given the 2D models are distorted, isn’t there an issue measuring polyp diameters from them? How was this dealt with?

I found figure 6 very interesting. In theory we should not be seeing any change in D across scale right?...if they are truly fractal. That’s kind of the point of fractals. So what’s going on here? Could it be that fractal dimension calcs are inherently constrained by the underlying resolution of models. So this just reflects out inability to maintain very high res models at larger spatial extents?

Minor Comments
Ln 124: Dp is not defined at this point.
Ln 186: what tool was used to get fractal dimension.
Ln 189: need to explain how it was validated a bit more.
Ln 220: 3D perimeter length is somewhat confusing. Perhaps something like perimeter from model?
Ln 237: “corals” rather than “outputs” right?
Ln 282-285: this sentence is confusing.
Ln 293+: I’m not sure its all the useful to be talking so specifically about individual colonies by name here.
Ln 319: R2 should be 0.18 according to the figure, not 0.8.
Ln 320: being significant.
Ln 401: surface fractal
Figure 4&5: you need to put the variable name key in the legend.

---

## Round 0.2 · Minor Revisions

Dear Dr. George,

The reviewers are happy with your revised ms and response to their comment in general and recommended your ms for publication after minor revision. I agree with their recommendation and am happy to inform you that your ms will be accepted for publication after minor revision. Please think about if you can include a parameter related to the dead area of the colony into your model to address the remaining concern of the second reviewer.

Thanks. Pei-Yuan Qian

Reviewer 1 ·

Basic reporting

The paper is well written, and with the revisions, particularly with the clarification of space-filling dimensions, it is much easier to follow.

Experimental design

Authors' research question is very well defined. Again, the clarification on the calculation of space-filling dimensions solved all my questions.

Validity of the findings

The results are very interesting, and I can now see how space-filling dimensions can capture more than just the surface area of coral colonies and integrate the measure of roughness/smoothness. It, however, makes me wonder if the fact that the metric additionally captures the presence of holes confounds the results, especially for losing species. Coral colonies can have dead or dying areas/patches for any reasons (unrelated to surface area and roughness/smoothness), so having such areas leading to lower values of surface space filling dimension, which then relate to an increased losing outcome, seems a bit circular to me. To me, the presence of holes itself sort of implies the losing state (more holes = more losing). I don't think it is serious enough to completely invalidate the results, but the fact that the space-filling dimension used in the study is affected by the presence of holes may need to be addressed in the manuscript. It leaves me wondering what would have happened if the study was conducted using colonies without any holes (i.e. competitions occurring only at the edges of the colonies).

Additional comments

Thanks again for all the revisions. The paper is very interesting.

Reviewer 3 ·

Basic reporting

No specific comments here.

Experimental design

The authors have done a very good job of addressing the concerns of myself and the other reviewers regarding the explanation of methods and analysis. The switch away from fractal dimension and towards space-filling terminlogy was a good one.

Really the only concern I have left is that regarding my comment that the amount of dead tissue on the colony could be related to the competitive success direclty....rather than via the space filling measure which is also affected by the presence of holes created when these areas of dead tissue were manually removed from the mesh. While I appreciate the addition of a figure to the supplementary material, I didn't find the provided argument very compelling largely because it didn't provid much context for the issue. That there are a few examples of corals that competed well and had dead tissue doesn't mean there might not be an overall effect here. I think a more convincing argument would perhaps be centered around clarifying that perhpas the number of holes was actually quite small...or their area relative to the colonies? Of course this is harder to make given the importance given to the effect of the holes on the spacd-filling dimension. So probably the best way to address this would simply be to include a metric related to the amount of dead tissue into the analysis and see if this does indeed have an effect. This could perhpas just be expressed as a percentage of the total colonly surface area, or something like that.

Validity of the findings

Aside from my concern about the amount of dead tissue on the colony being an important measure to consider in the analysis (as outlined above), I had no other concerns about the validity of the finding.

Additional comments

Thank you for your thorough efforts to address all comments.

---

## Round 0.3 · accepted · Accept

I am happy with the revised version and decide to accept it for publication.